# Assessment of Water Productivity Enhancement and Sustainability Potential of Different Resource Conservation Technologies: A Review in the Context of Pakistan

**Muhammad Adnan Shahid** [1,2], **Junaid Nawaz Chauhdary** [3,4,*], **Muhammad Usman** [5],
**Muhammad Uzair Qamar** [1,6] **and Abdul Shabbir** [1]

1   Department of Irrigation and Drainage, University of Agriculture, Faisalabad 38000, Pakistan;
    muhammad.shahid@uaf.edu.pk (M.A.S.); muhammad.uzair@uaf.edu.pk (M.U.Q.);
    abdulshabbir@uaf.edu.pk (A.S.)
2   Agricultural Remote Sensing Lab (ARSL), National Center of GIS and Space Applications (NCGSA),
    Faisalabad 38000, Pakistan
3   Water Management Research Centre, University of Agriculture, Faisalabad 38000, Pakistan
4   Department of Agricultural and Biological Engineering, Purdue University, West Lafayette, IN 47907, USA
5   Faculty of Natural Sciences, Institute of Earth Sciences and Geography, Martin Luther University Halle,
    06886 Halle (Saale), Germany; muhammad.usman@geo.uni-halle.de
6   Department of Civil Engineering, Schulich School of Engineering, University of Calgary,
    Calgary, AB T2N 1N4, Canada
*   Correspondence: junaid.nawaz@uaf.edu.pk; Tel.: +92-347-7196248

**Abstract:** Agriculture is the major economic sector in Asian countries and the majority of their population depends on it. In addition to the largest irrigation system in the Indus basin, Pakistan is suffering from water shortages that are affecting the overall crop production of the country. Different resource conservation technologies (RCTs) such as precision land leveling (PLL), raised bed planting (RBP), and different high-efficiency irrigation systems (HEISs) can be opted for better water productivity. In this study, the potential of these RCTs has been explored to enhance production and save irrigation water through their sustainable adoption. Based on studies by different researchers, water savings up to 47% and yield increases up to 35% have been reported under PLL, while water savings up to 50% and about 10–33% yield increases were observed under RBP. Similarly, under different HEISs, water savings up to 80% and yield increases up to 53% have been reported compared with crops sown under conventional farming. Based on the findings of the researchers regarding RCTs, these have been proved as progressive sowing techniques for better productivity under the limited available water scenario. The detailed review in this paper concludes that RCTs resulting in the improvement of gravity irrigation methods, viz., PLL and RBP, have a great potential of adoption and water productivity improvement at the regional scale in developing countries such as Pakistan, while high-cost HEISs can also be promoted at limited scale among progressive farmers for high-value agriculture.

**Keywords:** resource conservation practices; sustainable agriculture; precision land leveling (PLL); raised bed planting (RBP); high-efficiency irrigation systems (HEIS)

## 1. Introduction

The agriculture sector in Pakistan is a major contributor to its economy with a 19.3% share to Gross Domestic Production [1]. Moreover, 43.7% of the population of the country is directly engaged in this sector and 62% of their livelihood is dependent on it [2]. The current population of Pakistan is estimated to be 229 million and, according to some projections, it could increase to 338 million by 2050 [3]. This accelerating population creates challenges for the agriculture sector to produce more food to fulfill the food requirement of the growing population, especially in the situation when most of the population is being faced with

food insecurity [4]. The agriculture sector is, therefore, under pressure due to the threats of food insecurity, and the situation could be worse in the near future if some sustainable options are not adopted.

In Pakistan, more than 84% of the cultivable area is comprised of arid to semi-arid climatic zones, in which agriculture purely depends on irrigation water, and the irrigation water is responsible for producing more than 90% of the food of the country [1]. This situation highlights the importance of water security along with food security, as Pakistan is already facing water scarcity nowadays. This can be assessed from the statistics that the per capita water availability in Pakistan is below 1000 $m^3$, whereas, as per international standards, a country is said to be water-scarce if the per capita water availability goes below 850 $m^3$. The projected water availability in Pakistan will be 800 $m^3$ in 2025 [5]. According to a survey conducted by Asian Development Bank, there will be a 32% shortfall in water availability in Pakistan in 2025 and it will affect food production by 70 million tons [6]. Some important reasons for these present and projected water shortages are the escalating population, climate change impacts, poor surface water storage capacity, and poor performance of the irrigation system in terms of high conveyance and application losses.

The Indus Basin Irrigation System is a key source of freshwater in Pakistan, and it is responsible for providing, on average, 139 MAF of water annually. The conveyance losses during the delivery of water from the main source to the field are about half of the amount of this freshwater. About 30 MAF of water falls into sea without beneficial use due to the lack of storage capacity in the country [7]. As a result, water diversions in canals from rivers have decreased from 104 MAF to 98 MAF during 2012–2013, showing decreasing water availability. On a short-term basis, without going into major measures such as the construction of dams/barrages and the diversion/improvement of main canals, 47 MAF can be saved only by different measures in the tertiary irrigation system such as lining watercourses and the improvement in irrigation application efficiency. This reduction in conveyance losses and improvement of water application efficiency are the needs of the hour for sustainable agriculture in a water-scarce country such as Pakistan.

According to a public release by ADB (Asian Development Bank), Pakistan has the lowest efficiency among the countries in Asia at managing its water resources in terms of quality and quantity and further food production. An example of this poor performance can be seen in terms of the very low water productivity for wheat, i.e., 0.5 $kg/m^3$, as compared to 1.0 $kg/m^3$ in India and 1.5 $kg/m^3$ in California. Similarly, the water productivity for maize is only 0.3 $kg/m^3$ in Pakistan compared with the highest value of 2.7 $kg/m^3$ in Argentina [8]. In addition to poor performance, this also indicates a high potential for improvement in water productivity through the adoption of different Resource Conservation Technologies (RCTs).

Among various RCTs, Precision Land Leveling (PLL), Raised Bed Planting (RBP), and different types of High-Efficiency Irrigation Systems (HEISs) have proven their applicability globally to reduce water application losses and improve yield and, hence, water productivity. PLL, or more commonly known as LASER land leveling, ensures the reduction in deep percolation and evaporation losses due to uneven fields having depressions and undulations. RBP results in water savings due to the efficient application of water through furrows instead of flooding the whole field. Similarly, different types of HEISs provide about 80–90% application efficiencies by supplying water near to the plants' root zone in an exact amount. In addition to water savings, all these techniques result in better crop yields by providing a better crop stand under a uniform distribution of water. However, there may be a number of socio-economic, environmental, and technical factors, which may hamper the successful promotion and sustainable adoption of any of these RCTs in a region.

The adoption of these RCTs for improving water productivity is in dire need in Pakistan too, where a lot of research and promotional work has already been performed on these RCTs. However, keeping in view the abovementioned factors, an in-depth analysis is required to review the performance of these RCTs in Pakistan, as well as to identify different constraints and suitable guidelines linked with agroecological conditions of

different regions for their sustainable adoption. This paper presents a thorough review of the existing status of these RCTs regarding their adoption and research conducted on different aspects of these RCTs for their potential to improve crop and water productivity in Pakistan. The review and guidelines provided in this study may be greatly helpful for policymakers and the farmers for promoting and adopting the best suitable RCTs as per their local conditions.

## 2. RCTs to Improve Agricultural Water Productivity

To ensure future food security, it is highly important that the available water resources be precisely used to enhance water use efficiency and agricultural water productivity. This can be achieved by effectively adopting different RCTs or Water Conservation Techniques (WCTs) such as precision land leveling, raised bed planting, and different high-efficiency irrigation systems. In the following section, these RCTs have been thoroughly reviewed regarding different research aspects and their adoption status in Pakistan.

### 2.1. Precision Land Leveling

Precision Land Leveling (PLL) is a process of smoothing and grading the land to a plan surface or uniform surface at an angle up to the precision of ±2 cm. This technology is not new in the area of precision agriculture. It has been proven by many scientists in the past to be an efficient tool for achieving a high degree of precision. Leveled fields through LASER technology help to improve the water application efficiency and enhance crop yields due to uniform water distribution.

Johnson [9] investigated the physical and economic benefits of precision land leveling in Pakistan by comparing two sets of fields, i.e., precision-leveled and those leveled using traditional techniques. Overall, a 35% improvement in yield was reported under a LASER-leveled field with 2274.7 kg/ha of production compared with 1684.1 kg/ha of production under a conventionally leveled field. The water saving was 16% under precision land leveling compared with a conventionally leveled field.

Precision land leveling acts as a catalyst and proves itself to be a valuable technology by increasing the efficiency of other associated inputs. That is why LASER land leveling has been a success story in Pakistan as with many other countries. In this regard, a study carried out by the Agriculture Department during 2008 revealed various benefits of LASER land leveling including an irrigation time saving of 25–32% with a yield improvement of 11–13%. Farmers were found convinced that LASER-leveled fields provided a better crop stand, uniform moisture availability, and enhanced fertilizer use efficiency, resulting in ultimate benefits of water savings and yield increases. It was reported by impact evaluation that the adoption of LASER land leveling resulted in increasing irrigated areas from 34.5% to 42% and reducing farm culturable wasteland by 2.10% [10].

Abdullaev [11] compared PLL with nonleveled fields and reported that PLL reduced deep percolation losses by 8%. In addition, the loss through runoff was reduced to 24%. The average annual net income from the field under PLL was 22% higher than that under unleveled fields. Jat [12] conducted a study to establish the understanding of how wheat yield and water use efficiency behaved under PLL. The results showed that bed planting under PLL produced a 16.6% higher wheat yield and saved irrigation water by 50% compared to traditional practices (traditional land leveling with flat planting). In another study, Jat [13] discussed the rice-wheat system of Punjab to compare the impact of PLL in comparison to conventional flat sowing. The PLL improved water productivity by 7.4% for the rice-wheat system and saved irrigation water by 12–14% for rice and 10–13% for wheat. By improving the crop yield and saving irrigation water, PLL enhanced profits by US\$ 113 ha$^{-1}$ in the first year and US\$ 175 ha$^{-1}$ in the second year.

Kaur [14] investigated the environmental and economic benefits of PLL using farm-level information from the Moga district of Punjab. The researchers reported that PLL saved irrigation water and energy by 24%. Rice produced 4.25% more yield under PLL as compared to conventional sowing practices. The average reduction in irrigation cost

was 44% and the increase in water productivity was 39% under PLL for rice. Rizwan [15] reported that PLL was performed at six sites (Khurrianwala, Killianwala, Mungi, Dijkot, Khikhi, and Shahkot) in the command area of the Lower Chenab Canal system. It was reported that water savings under PLL were 18.7% for wheat, 19.9% for cotton, and 22.4% for rice as compared to that under conventional leveling. The average water saving under PLL was recorded as 20.3%.

Keeping in view the reported success and benefits, the technology is being further promoted in Pakistan under different government projects by providing LASER leveling facilities through district-level departments of OFWM, as well as by providing LASER machinery to the farmers on subsidies. Table 1 provides a summary of the results regarding the benefits of PLL as reported under different studies here in the review.

**Table 1.** Water saving, yield increase, and water productivity improvement under PLL.

| Sr. | Water Saving (%) | Yield Increase (%) | Increase in Water Productivity (%) | Increase in Income | Remarks | Reference |
|---|---|---|---|---|---|---|
| 1 | 46.8% | 35.3% | 75.4% | - | Crop sown under PLL with bed planting | [9] |
| 2 | 30% | - | - | 22% | - | [11] |
| 3 | 25–32% | 11–13% | 10–44% | - | - | [10] |
| 4 | 50% | 16.6% | 33.2% | - | Wheat sown under bed planting with PLL | [12] |
| 5 | Rice: 12–14% Wheat: 10–13% | - | 7.4% | US$ 144 ha$^{-1}$ | The data were collected in rice-wheat system of Punjab, India | [13] |
| 6 | 24% | 4.25% | 39% | - | - | [14] |
| 7 | Wheat: 18% Cotton: 16.8% Rice: 22.4% Average: 20.3% | - | - | - | - | [15] |

### 2.2. Raised Bed Planting

Raised bed planting is an improved surface irrigation technique in which water is applied only in the furrows, causing significant water savings. This technique has many advantages over conventional surface irrigation. Sayre [16] reported less lodging of wheat sown in beds that ultimately increased the final grain yield. Other studies have shown better fertilizer use efficiency, less weed infestation, and saving a reasonable amount of irrigation water (by almost 35% to 45%) under bed planting [17], thus resulting in enhanced water productivity. Better yield has also been reported even with the use of a lower quantity of seed [18]. Having been encouraged by the advantages of bed planting and in view of the necessity of adopting this improved water conservation technique, On Farm Water Management (Punjab) introduced raised bed technology in the rice area of Gujranwala, Sialkot, and Sheikhupura using an Indian-made bed planter cum seed drill. However, PARC [19] reported problems in the wheat planting on beds using this machine and indicated less uniformity during the placement of fertilizer and seed, which was identified as a major issue for further investigations.

The Water Management Research Centre (WMRC) at the University of Agriculture, Faisalabad conducted a series of research studies on the geometry of bed planting for wheat production regarding the enhancement in water and crop productivity. Based on the comprehensive reassertion outcomes, WMRC developed a bed planting machine that can sow wheat crop by making two beds and three furrows in a single operation. There

is a provision for farmers to sow sugarcane as an inter-crop in standing wheat under the recommended geometry of bed planting for wheat [20].

In the subsequent years, intensive research and outreach activities were carried out to check different aspects of this technology regarding its sustainable adoption. Ahmad [21] carried out a study to determine the impact of raised bed planting on yield and the lodging of wheat, as well as on water savings as compared to wheat under conventional methods. The results indicated the minimum impact of lodging in raised bed planting (20.5%) as compared to flat sowing (34.6%). It was reported that reduced lodging in raised beds was due to the drainage of excessive rainwater through furrows and the strong crop stand on beds. Water savings at different locations were observed in the range of 40–50% for bed planting, while the average yield increase under raised bed planting was recorded as 11.23% and 16.20% for nonlodged and lodged fields, respectively.

The raised bed planting technology was transferred to the farmers in the area of T.T. Singh from 2004 to 2007 (Figure 1) under the UAF Technology Transfer Program for its testing and demonstration at farmers' fields and fine tuning after receiving feedback from them. The success results were shared with farmers in addition to receiving their feedback regarding any problems/constraints. Average yield increases of 10–15% were observed for wheat and 10–25% for maize, while water savings were in the range of 40–45% [22]. The results regarding the water saving and yield increase for raised bed planting of wheat are presented in Figure 2 for three consecutive seasons, whereas the benefit of raised bed planting in terms of resistance against lodging is highlighted in Figure 3. The comparison of irrigation under bed planting and flat sowing are shown in Figure 4.

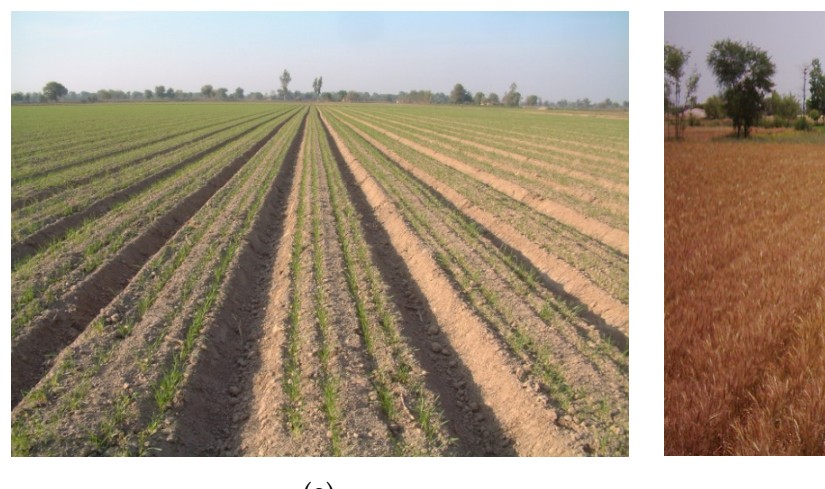 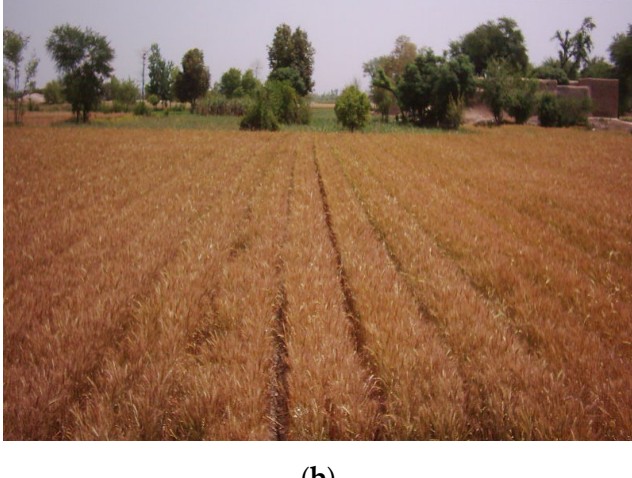

(**a**)　　　　　　　　　　　　　　　　　　　　　　　　　　　　　　　　(**b**)

**Figure 1.** Different phases of wheat planting on beds at farmers' fields under UAF technology transfer program. (**a**) Wheat right after germination on beds; (**b**) wheat at full maturity right before harvesting.

During the promotion of raised bed planting, a misconception in the farming community was observed where this method of sowing may provide water savings due to under-irrigations, which will finally affect the crop yield. To remove this misconception, it was very important to conduct research and dissemination to the farmers regarding the comparison of water/irrigation requirements and applications under both raised bed planting and conventional methods. This comparison of on-farm water requirements and applications revealed water losses for the first irrigation in both methods with the highest water losses for flat sowing. In all other irrigations, flat sowing showed water losses, whereas the irrigation applications in bed-planted treatments were found almost according to the need. In the fourth and final irrigation, some under-irrigation in bed planting was seen, but it did not affect the yield, as the crop was on the maturity stage and may bear some stress without compromising on yield, a concept of deficit irrigation. The average

water saving under raised bed planting in comparison to flat sowing was found to be 50% with an average water productivity of 1.31 kg/m$^3$ against 0.56 kg/m$^3$ for flat sowing [23].

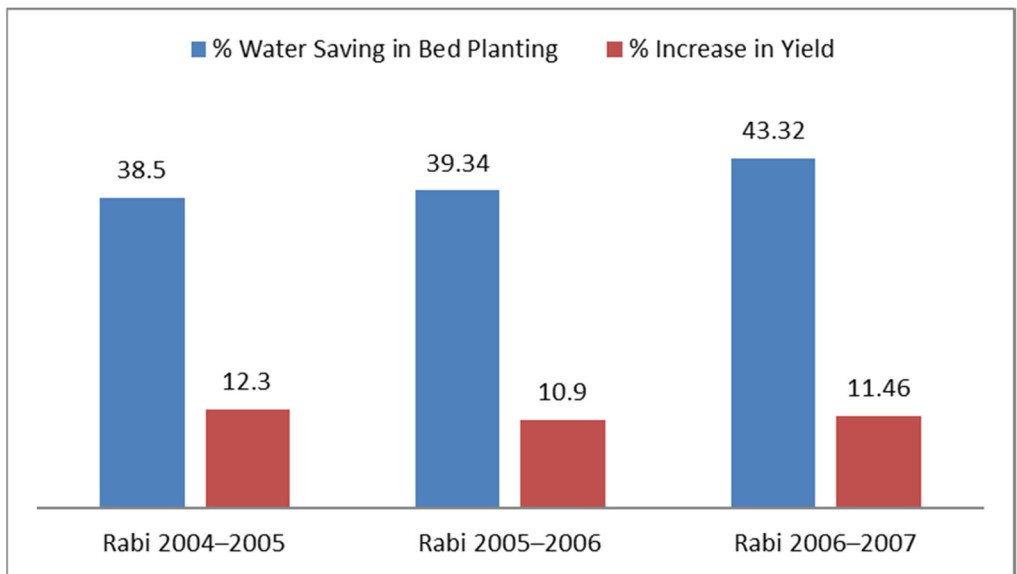

**Figure 2.** Wheat planting on beds at farmers' fields under UAF technology transfer program.

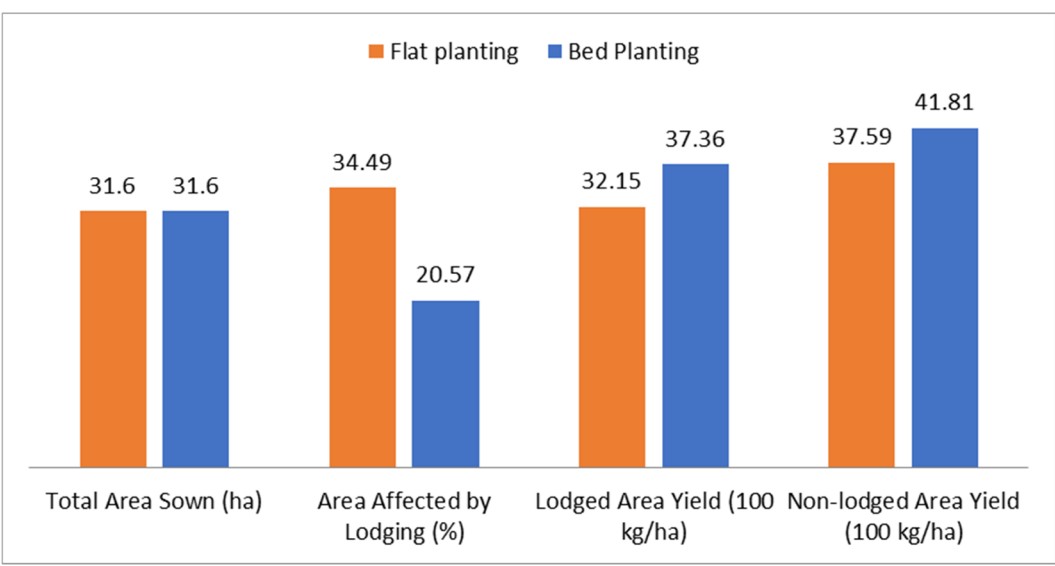

**Figure 3.** Benefits of raised bed planting to provide resistance against lodging.

Shahid [24] investigated the optimum number of irrigations, essential for the better productivity of wheat under raised bed planting. The investigation included flat sowing and bed planting with irrigation applications at 50% Management Allowed Depletion (MAD). One treatment was under bed planting with irrigation on the same day as under flat sowing, and the moisture depleted more than 50% MAD. The number of irrigations were higher (five irrigations) under raised bed planting, whereas four irrigations were applied under flat sowing. It was observed that the grain yield and water productivity were higher under raised bed planting with four irrigations with a 47.43% water saving. Under raised bed treatment with five irrigations, the water saving dropped from 47.43% to 35.74%, while no significant difference in yield was observed. Therefore, it was recommended to not apply extra irrigation under raised bed planting for better crop and water productivity.

Chauhdary [25] conducted a two-year study on wheat to investigate the best sowing method among (1) broadcasting, (2) drill sowing, and (3) bed planting and optimum seed

rate among (1) 100 kg/ha, (2) 130 kg/ha, and (3) 160 kg/ha for better water and crop productivity. It was found that a 160 kg/ha seed rate produced the highest grain yield (4117.1 kg/ha) and water productivity (1.8361 kg/m$^3$). It was also reported that bed planting saved 35% of irrigation water as compared to that under flat sowing (broadcasting and drill sowing). The wheat yield was higher under bed planting by 11% and 3.3% as compared to those under broadcasting and drill sowing, respectively. Similarly, water productivity was higher under bed planting by 42.7% and 8.4% as compared to those under broadcasting and drill sowing, respectively. Bakhsh et al. (2018) verified these findings by reporting 31.5%, 38.7%, and 42.6% water savings and 90.6%, 82%, and 108% better water productivities under raised bed planting for rice, cotton, and wheat, respectively.

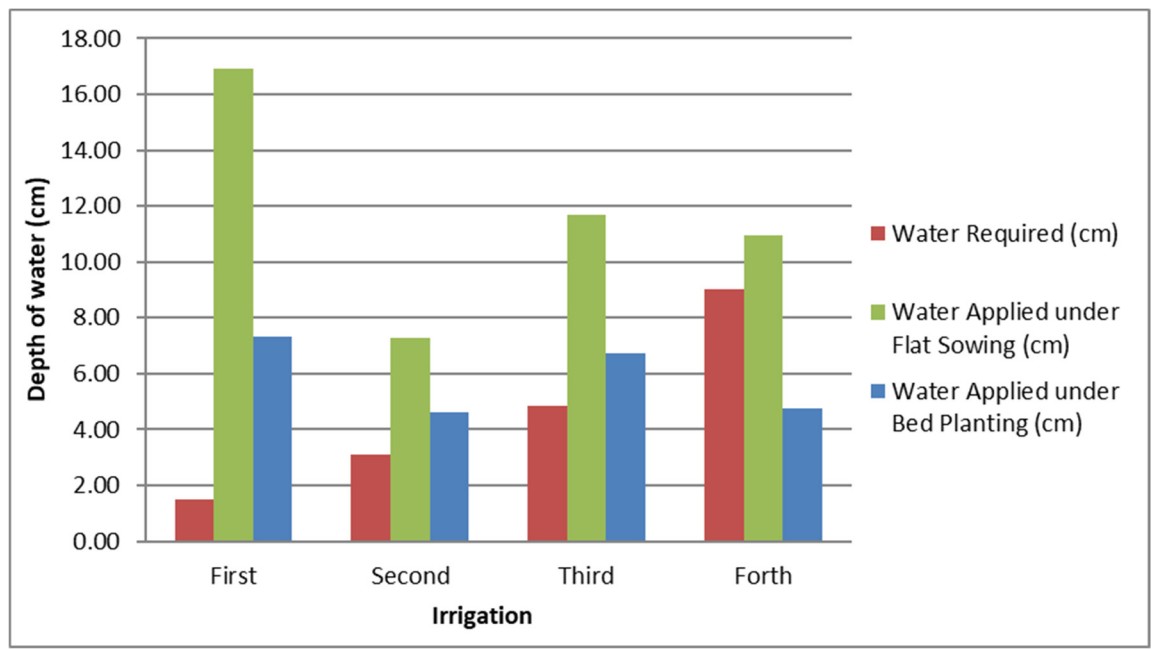

**Figure 4.** Comparison of irrigation requirements and applications [23].

In addition to research on different aspects of raised bed planting and the initial technology transfer program in T.T. Singh, the technology has been continuously promoted at farmers' fields in Faisalabad and T.T. Singh districts under different projects for the last eight years. Ahmad [26,27] reported dissemination activities of raised bed planting in rice-wheat cropping areas of Faisalabad. Based on three years of trails at farmers' fields from 2007 to 2010 under the On-Farm Research & Development Component Project, an average water saving of 48.26% was recorded in the bed planting of wheat in comparison to flat sowing, with an average yield increase of 21.44%. Another benefit of wheat bed planting was introduced by promoting the intercropping of sugarcane in furrows of bed-planted wheat in the month of February. It resulted in considerable water savings as compared to conventional sugarcane planting in May, while the yield under this approach was recorded as 79,000 tons as compared to 66,000 tons in the case of the conventional method. Moreover, it helped to avoid competition between sugarcane and wheat regarding their cropping areas being in the same season. For rice in the Kharif season, the bed planting of rice was observed to give an average water saving of 30.15% and yield increase of 32.67% in comparison to conventional sowing.

Similarly, in the cotton/maize-wheat area of T.T. Singh, raised bed planting was promoted for maize and cotton under the OFR&D project and technology transfer project funded by the UAF Endowment Fund. About 1000 acres of cotton bed planting and 200 acres of maize bed planting were performed each year [27]. The average yield increase under maize bed planting was observed as 18.75% with an average water saving of 32.26% in comparison to ridge sowing. The yield increase for cotton bed planting was recorded

as 11% on average with an average water saving in the range of 30–40% in comparison to conventional ridge sowing. Raised bed technology has also been promoted recently under the OFR&D project, as well as under the USAid-funded project of Watershed Rehabilitation. In the R&D project, the major focus was on planting about 1000 acres each year using furrow-bed technology by providing machine and other services at the farm. In the USAid-funded project, bed planting was promoted by establishing demo plots, as well as by distributing machines among farmers on subsidies with the condition that they will ensure bed planting on about 100 acres each. In addition, the modification in the design of a bed planter, to improve it from manual operation to pneumatic operation, is in process by the researchers of the Department of Irrigation and Drainage, UAF. A summary of the review regarding raised bed planting is given in Table 2.

**Table 2.** Water saving, yield increase, and water productivity improvement under Raised Bed planting.

| Sr. | Water Saving (%) | Yield Increase (%) | Increase in Water Productivity (%) | Remarks | Reference |
|---|---|---|---|---|---|
| 1 | 40–50% | 11.23% for lodged field 16.20% for nonlodged field | 22–28% for lodged field 32–34% for nonlodged field | Research was conducted on wheat in T.T. Singh | [21] |
| 2 | 40–45% average for both wheat and maize | 10–15% for wheat 10–25% for maize | | Research was conducted at T.T. Singh for three consecutive seasons | [22] |
| 3 | 50% | - | 133% | | [23] |
| 4 | - | 12.5% | - | Research was conducted on wheat | [28] |
| 5 | 47.43% | - | 144% | | [24] |
| 6 | 48.26% for wheat 30.15% for rice 39.2% as average | 21.44% for wheat 32.67% for rice 27% as average | 117% for wheat 108% for rice 112.5% as average | Multi-seasonal study was conducted at multiple sites of Punjab, Pakistan | [26] |
| 7 | 32.26% for maize 30–40% % for cotton | 18.75% for maize 11% for cotton 14.9% as average | 58% for maize 28–37% for cotton | Multi-seasonal study was conducted at multiple sites of Punjab, Pakistan | [27] |
| 8 | 35% | 11% | 42.7% | - | [25] |
| 9 | 42.6% for Wheat 38.7% for Cotton 31.5% for Rice 37.6% as average | 19.9% for Wheat 12.1% for Cotton 22.8% for Rice 18.3% as average | 108% for Wheat 82% for Cotton 90.6% for Rice 95.5% as average | - | [29] |
| 10 | 47.2% for Wheat 32.3% for Cotton 27.1% for Rice 35.7% as average | 8–16.6% for Wheat 8.6–14.5% for Cotton 25.1–28.1% for Rice 13.9–19.7% as average | 16.9–35.1% for Wheat 26.6–44.9% for Cotton 92.6–104% for Rice 45.4–61.3% as average | - | [15] |

### 2.3. High-Efficiency Irrigation Systems

High-Efficiency Irrigation Systems (HEISs) are pressurized irrigation techniques, which have been found water- and nutrient-efficient and the most appropriate option to address various water losses and crop production issues. Different types of HEISs include drip, bubbler, conventional sprinkler, rain-gun, and center pivot, which use pipes to convey water from the source to the points of application. Drip irrigation is the most efficient technology among the HEISs that can apply water, fertilizer, and nutrients effi-

ciently. Under drip irrigation, the regular availability of moisture and nutrients as per plant needs throughout the crop period facilitates maximizing crop productivity. Therefore, it has become the most valued innovation, which optimizes the use of water and fertilizers by enhancing the irrigation efficiency as much as 95 percent. However, common problems with all kinds of HEIS are their high initial cost and complexities in their operation and maintenance. These problems make their massive adoption a challenge, although the techniques are not very new and their benefits have been investigated and reported for the last 30–40 years. The drip irrigation should be adopted to increase water-use efficiency while maintaining reasonable yields compared to furrow irrigation [30]. It has further been reported that subsurface drip irrigation offers the highest efficiency for applying soil moisture and plant nutrients uniformly.

In a study [31], researchers compared drip irrigation with basin irrigation for applications of different amounts of water (Epan: 0.4, 0.3, and 0.2) and nitrogen (100%, 75%, and 50% of full dose) and reported a 32% higher cotton yield and 26% better water use efficiency (WUE) under drip irrigation corresponding to each level of water and nitrogen. Effects of two irrigation methods (surface irrigation and drip irrigation) and three plant spacings (10 cm, 20 cm, and 30 cm) were evaluated by Dilbaugh [32] for the production of seed cotton. The results revealed a better production of yield and the yield components of seed cotton under drip irrigation over conventional furrow irrigation with 7.9 kg/ha/mm WUE and a 53.3% water saving. The narrow-spacing crop produced a higher yield than widely spaced lines. Singadhupe [33] studied the response of sugarcane under drip irrigation with different irrigation intervals (2 days, 3 days, and 4 days). Sugarcane produced 20%, 16%, and 13% higher yields corresponding to each interval, respectively, compared with that under flood irrigation. Similarly, the highest water productivity (108% more than that of flooded crop) was achieved by a 2-day interval under drip irrigation. Drip fertigation proved to be effective for apparent crop recovery and improving physiological and agronomic efficiencies than furrow irrigation. The response of corn, sown under drip at three different cropping zones of Punjab, Pakistan, is shown in Figures 5 and 6 in terms of water applied and crop yield, respectively. Chauhdary conducted a series of research experiments [34–36] to study the impact of different irrigation frequencies (daily irrigation, 3rd day irrigation, and 5th day irrigation), fertigation rates (50%, 75%, and 100% RDF), and fertigation compounds (imported and indigenous) on maize crop. The amount of irrigation was calculated based on pan data and the fertilizer amount was calculated based on reference standards @ N:P:K = 250:125:125 kg/ha. Different techniques were adopted in each study. The highest economic yield was obtained under 100% RDF of indigenously developed acidic fertilizer with daily irrigation applications through drip. Using the modeling technique, the results were extrapolated, and it was reported that beyond 150% RDF, the feasibility of drip for crop production starts to decline [37]. El-Rahman [38] conducted a field experiment in Egypt to study the system efficiency of single-lined and multi-lined drip irrigation with two flow rates, i.e., 4 L/h and 8 L/h, for wheat crop. The biological yield (straw yield and grain yield) of wheat was increased with multi-lined drip irrigation with less flow rate, viz., 4 L/h. The total applied water for wheat was 2087 $m^3$. The average yield and the water use efficiencies of different wheat varieties (Giza 69, Sakha 8, and Giza 7) were (2085 kg, 2120 kg, and 2145 kg) and (1.13 kg/$m^3$, 1.20 kg/$m^3$, and 1.17 kg/$m^3$), respectively. Anjum [39] investigated different irrigation management practices including different irrigation systems (drip irrigation and bed-furrow irrigation), different irrigation intervals (2-, 4-, and 6-day intervals), and different water quality levels (poor-, marginal-, and good-quality water). When compared with drip irrigation, the crop under bed planting performed better in terms of plant height (1% more), dry matter (5.8% more), and grain yield (21.9% more). The results of the study showed that irrigation frequency had a quadratic relation with dry matter weight, grain yield, harvest index, and plant height. Two- and six-day irrigation intervals showed better performance regarding all these parameters as compared to the four-day interval. The grain yield was 5.33, 6.38, and 7.5 t/ha under poor-, marginal-, and good-quality water, respectively. Good-quality water also

upgraded the plant dry matter weight by 11.7%. Bakhsh [40] reported that drip irrigation produced a maximum WP of 2.26 kg/m$^3$ for wheat with a 98% application efficiency. The water saving under drip irrigation was recorded as 40% when compared with that under conventional irrigation.

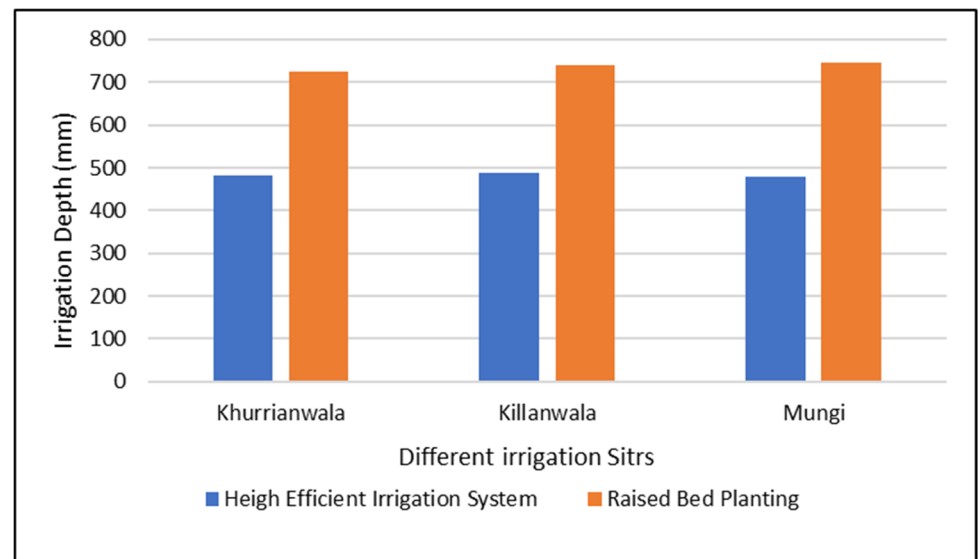

**Figure 5.** Comparison of irrigation depth (mm) for maize crop, sown at different sites [15].

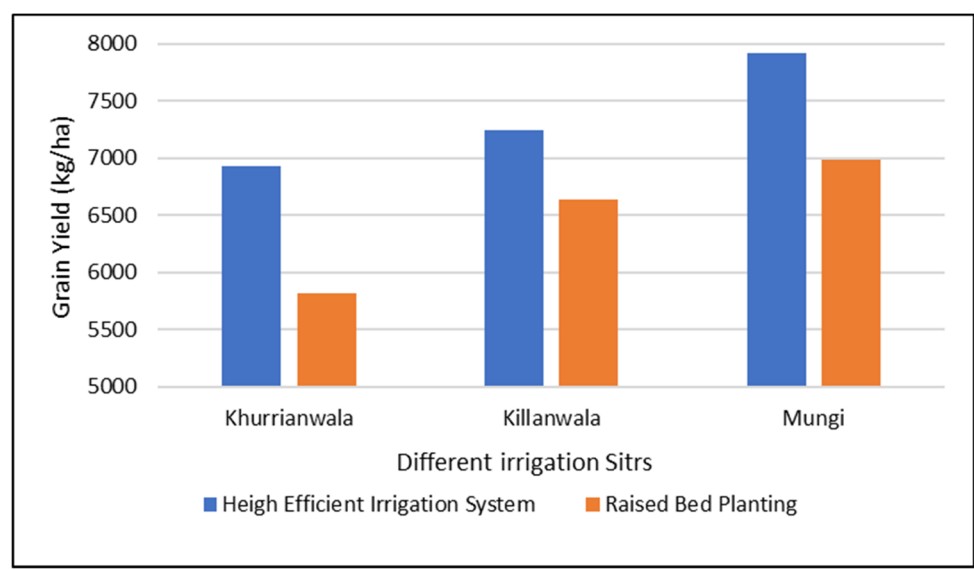

**Figure 6.** Comparison of grain yield (kg/ha) of maize, sown at different sites [15].

The numerous benefits of drip irrigation for production of crops have been discussed but these benefits could be huge if it is used for high value agriculture like production of vegetables. Yohannes [41] conducted experiment in two consecutive years i.e., 1992 and 1993 to compare the tomato response under drip irrigation and surface (furrow) irrigation methods; higher tomato yields were reported under drip irrigation. Similarly, plant height and fruit size were better under drip applications and also the irrigation application efficiency and water use efficiency (WUE) were highest in comparison to those under furrow irrigation. Different irrigation schemes were also examined in an on-farm research study comprising surface irrigation and low cost drip irrigation [42]. The impact was assessed on the vegetable "English giant rape (Brassica napus)" under both irrigation (surface and drip) and fertilizer (without fertilizer, regular solid fertilizer and liquid fertilizer) schemes. The

vegetable yield (8.5 t/ha) was 9% more than that under surface irrigation (7.8 t/ha), whereas drip system coupled with fertigation produced (10.8 kg/m$^3$) three and half times better water productivity, compared with surface irrigation (2.4 kg/m$^3$). Therefore, drip irrigation was recommended for economic production of vegetables. In another research, the saline water effects were studied on tomato, sown under drip irrigation in China [43]. Results showed that saline water (1.1–4.9 dS/m) did not show significantly negative impact on crop yield, WUE and IWUE. Moreover, salts were not accumulated in upper soil profile even after three years of study. Therefore, it was concluded that if good quality water is not available, then less saline water (2.2–4.9 dS/m) could also be used for irrigation under drip irrigation for crop production. The tomato was sown under drip with four irrigation levels (60% Et, 80% ET, 100% ET and 120% ET) [44]. The volume of water applied per plant under each level was 52.720L, 61.451L, 69.607L and 79.524 L, respectively with 91% system efficiency and 94% distribution efficiency. The highest water productivity was obtained with 60% ET. In this study, farm yard manure (FYM), vermi-composed (VC) and chemical fertilizer (CF) were also compared and it was found that water use efficiency was highest under FYM (10.4 t/ha/mm) than under CF (9.12 t/ha/mm). The comparative analysis showed that highest net return (Rs. 363,016/ha) with 5.9 BCR was acquired under drip with 100% ET water applications coupled with FYM. Second effective combination was 100% ET with the applications of VC with net income of Rs. 3, 45,808/ha and 4.98 BCR. Both combinations were recommended depending upon the availability of fertilizer compounds. Singh [45] studied feasibility of drip system based on net present worth (NPW) for capsicum. The treatments were included different irrigation levels (0.6, 0.8 and 1 of ET) with and without mulch. The amount of water applied was 249 mm, 332 mm and 415 mm under drip without mulch at 0.6 ET, 0.8 ET and 1 ET, respectively. Overall, water applications under mulch were less corresponding to same levels of ET under without mulch. 100% ET with mulch applications produced highest yield followed by same amount of water without mulch, while, the highest NPW was obtained by 100% ET (Rs. 309,734.90) and lowest was in case 0.6 ET with mulch (Rs.144, 172.24) applications. Overall, NPW was positive for all treatments. In another field study, efficiency of drip irrigation system for water and fertilizer use was examined in brinjal crop for different water regimes including 50%, 75% and 100% of EPan along with three fertigation levels, viz. 75%, 100% and 125% of full NPK dose, through drip irrigation [46]. The highest yield (42.33 t/ha), water and fertilizer use efficiencies were reported under 75% EPan irrigation applications along-with 75% NPK. Also, highest BCR was reported at 75% of EPan with 75% NPK in two consecutive years (2.9 and 2.5, respectively) from 2007 to 2008. Sezen [47] studied water use efficiency of pepper, sown under drip irrigation and its effect on the pepper yield against three irrigation intervals and three irrigation levels. Both irrigation intervals and irrigation levels passed significant impact not only on pepper yield, but also on fruit quality. The water productivity varied from 7.8 kg/m$^3$ to 6.0 kg/m$^3$ with maximum and minimum values under IF1DI1 and IF3DI3, respectively. Summary of different reviews about high efficiency irrigation systems is given in Table 3.

**Table 3.** Water saving, yield increase, and water productivity improvement under high-efficiency irrigation systems.

| Sr. | Water Saving (%) | Yield Increase (%) | Increase in Water Productivity (%) | Remarks | Reference |
|---|---|---|---|---|---|
| 1 | 44% | 53% | 120% | The study was on tomato crop under drip with mulch | [48] |
| 2 | 26% | 32% | 123% | The study was conducted on cotton | [31] |
| 3 | - | 8.9% | 350% | The study was conducted on vegetables | [42] |

**Table 3.** *Cont.*

| Sr. | Water Saving (%) | Yield Increase (%) | Increase in Water Productivity (%) | Remarks | Reference |
|---|---|---|---|---|---|
| 4 | 26–29% | 16% | 55–61% | The study was conducted on cane under drip in comparison to that under flat sowing | [33] |
| 5 | 53% | - | - | Cotton under drip irrigation | [32] |
| 6 | 43% | - | - | The study was conducted on maize under drip in intercropping-based system | [49] |
| 7 | 40% | 39% | 131% | The multi-seasonal study was conducted on wheat under drip in comparison to that under flat sowing | [29] |
| 8 | 48% | 39% | 118% | The study was conducted on maize under drip in comparison to that under conventional irrigation | [37] |
| 9 | 60–80% | 30–40% | 50% | The study was conducted on different crops in different cropping zones of LCC command area, Pakistan | [15] |

## 3. Discussion and Conclusions

In this paper, a detailed review regarding the benefits and scope of different RCTs was performed by citing individual studies of various researchers. The majority of the investigations by various researchers concluded that the RCTs including precision land leveling (PLL), raised bed planting (RBP), and high-efficiency irrigation systems (HEIS) have the potential to save irrigation water and enhance crop yield. Research studies showed up to 47% water savings and 35% yield increases under PLL, 30–50% water savings and 10–33% yield increases under RBP, and up to 80% water savings and 53% yield increases under HEIS compared with crops sown under conventional farming. From the detailed review, it is concluded that PLL and RBP have great potentials of adoption and water productivity improvement at the regional scale in developing countries such as Pakistan, while the high-cost HEIS may also be promoted in selective areas among the progressive farmers for high-value agriculture. In this regard, two important factors may be the soil type and the cropping patterns to decide on the potential areas for the sustainable adoption of different RCTs.

Figure 7 presents the spatial distribution of different soil types in the Punjab province of Pakistan based on digital soil map/data of the world developed by the Food and Agriculture Organization [50]. From the figure, it can be seen that major parts of the province, especially the upper and central Punjab, possess the loamy to sandy loam type of soil, which is considered suitable for any kind of gravity irrigation.

However, the Thal region in the central western part of the province and the major area in the south-eastern part of the province comprise sandy soils, while a thin belt on the south-western side and some patches in the northern part of the province contain sandy clay loam soils. All these areas having sandy clay loam to sandy soils possess great scope for the adoption of HEIS, including drip irrigation, rain-gun, and center-pivot-type sprinkler systems.

Figure 8 presents the distribution of major cropping zones in Punjab, Pakistan [51,52], based on MODIS-based land use land cover and major crop types mapping from 2005–2006 to 2015–2016.

Figure 8 shows that the south Punjab region is mainly considered as the cotton zone, having cotton as the major crop in Kharif and wheat as the major crop in the winter or Rabi season supported by vegetables in tunnels. The central and eastern part of this region from

Muzaffargarh to Bahawalpur may have great potential for the promotion and successful adoption of drip irrigation for the cotton crop, as well as for vegetables grown in the tunnels. The loamy sand type of the soil in this region (Figure 7) further advocates the potential for successful adoption of HEIS such as drip irrigation.

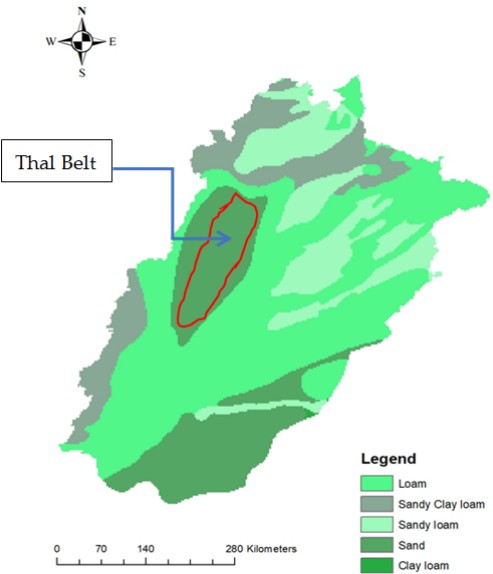

**Figure 7.** Soil texture classification for Punjab, Pakistan, extracted from Soil Map of the World [50].

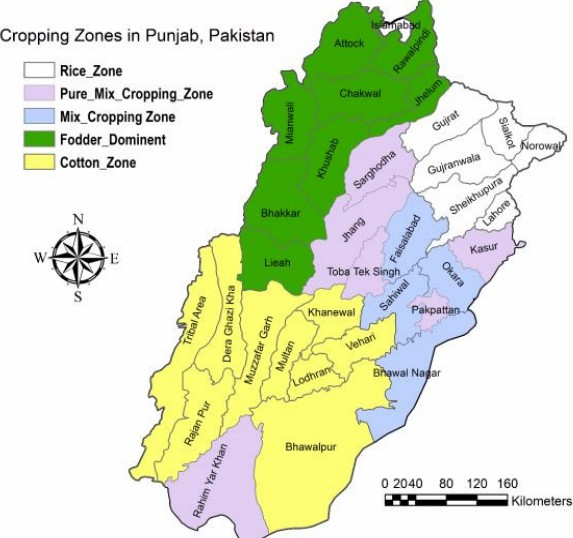

**Figure 8.** Cropping zones in Punjab, Pakistan [51,52].

The north-west region of the Punjab is mainly comprised of the fodder-dominant zone, where different types of fodder including Rhodes grass are grown on vast areas. Moreover, the Layyah, Bhakkar, and Khushab districts in the southern part of this zone are also famous for growing chickpea in the vast sand dunes of Thal. These chickpea areas are mostly rainfed or irrigated with pressurized high-efficiency irrigation systems such as drip and sprinklers. However, keeping in view the undulating topography, rain-gun sprinklers may be considered more feasible as compared to drip for this region, whereas the center-pivot-type sprinklers may be considered most feasible in other parts of the fodder-dominant cropping zone, where different types of fodders or grasses are grown.

For other parts of the province comprising mixed-crop and rice-wheat zones, the improved gravity irrigation methods such as Raised Bed Planting (RBP) coupled with Precision Land Leveling (PLL) should be promoted, as the soil types in these regions are

also sandy-loam to clay type, while the sufficient water supplies are also available under canal-irrigated agriculture supplemented with nominal groundwater resources. Thus, the promotion of pressurized irrigation systems in these regions has not been a great success up to now; the improved gravity irrigation techniques may help in achieving sustainable agriculture with improved water productivity in these regions.

Finally, based on careful integration of the information provided in Figures 7 and 8 as discussed above, the following site-specific guidelines are recommended for the successful adoption of different RCTs and sustainable agriculture:

- In rice-wheat and mixed-crop zones of upper and central Punjab and similar areas in other provinces, an improved gravity irrigation method of raised bed planting integrated with precision land leveling should be promoted, which may help achieve water savings up to 50% and yield increases of about 25–35% on average.
- In the cotton-wheat zone of south Punjab and similar soil and cropping zones in other provinces, drip irrigation may be successfully adopted for cotton crop and fruit orchards, as well as for different types of vegetables and fruits in tunnels. This will help achieve more than 80% water savings and up to 50% yield increases as compared to conventional gravity irrigation methods.
- Center-pivot type sprinklers are the best option for northern fodder-dominant zones of Punjab, while rain-gun-type sprinkler irrigation may be considered a viable option for the Thal region in the southern fodder-dominant zone; both sprinkler types have the potential to provide up to 80% water savings and 50% yield increases.

**Author Contributions:** J.N.C. and M.A.S. equally contributed to the technology review and manuscript drafting, in addition to the "discussion and conclusion" section by M.A.S.; M.U. and M.U.Q. developed GIS-based cropping zones of Punjab; A.S. helped the first and corresponding authors in interpreting and correlating the review outcomes to different soil and cropping zones. All authors have read and agreed to the published version of the manuscript.

**Funding:** This research received no external funding

**Institutional Review Board Statement:** Not applicable.

**Informed Consent Statement:** Not applicable.

**Conflicts of Interest:** The authors declare no conflict of interests.

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
