# Peer review of "Assessment of Water Productivity Enhancement and Sustainability Potential of Different Resource Conservation Technologies: A Review in the Context of Pakistan"

_agriculture, doi:10.3390/agriculture12071058_

Round 1

Reviewer 1 Report

In this study, the potential of these RCTs has been explored to enhance production and save irrigation water through their sustainable adoption. Based on studies by different researchers, water saving upto 47% and yield increase upto 35% have been reported under PLL, while water saving upto 50% and about 10-33% yield increase were observed under RBP. Similarly, under different HEIS, water saving upto 80% and yield increase upto 53% have been reported compared with crops sown under conventional farming. The reviewer thinks the topic discussed in this paper is very important, which is of great significance for agriculture research. This reviewer sees that a minor revision will be needed. Here are the main comments for the revision.

(1) A key problem in this paper is the lack of introduction to the applicability of the proposed method.

(2) The introduction section lacks some important references, such as:

Centrifugal model test on a riverine landslide in the Three Gorges Reservoir induced by rainfall and water level fluctuation; Prediction of landslide displacement with step-like behavior based on multialgorithm optimization and a support vector regression model.

(3) Some Figs are unclear and unexplained. For example, figure 7 and figure 8 are not clear.

(4) The conclusion is not concise and innovative. I believe that the Authors should try to interpret and explain more clearly their results. Some key quantitative conclusions should be supplemented.

(5) In this study, the authors must improve the statements about what is new in their study and what are the contributions to the developments of agriculture.

(6) The text is not clear. The author is suggested to check the professional vocabulary, grammar and spelling of the full text.

Author Response

This manuscript is a resubmission of an earlier submission. The following is a list of the peer review reports and author responses from that submission.